# Contamination Level in Geo-Accumulation Index of River Sediments at Artisanal and Small-Scale Gold Mining Area in Gorontalo Province, Indonesia

**DOI:** 10.3390/ijerph19106094

**Published:** 2022-05-17

**Authors:** Satomi Kimijima, Masayuki Sakakibara, Sri Manovita Pateda, Koichiro Sera

**Affiliations:** 1Graduate School of Science &Engineering, Ehime University, 2-5 Bunkyo-cho, Matsuyama 790-8577, Japan; sakaki@chikyu.ac.jp (M.S.); manovita.pateda@gmail.com (S.M.P.); 2Department of Environmental Health, Faculty of Public Health, Hasanuddin University, Jalan Perintis Kemerdekaanno No. 94, Tamalanrea, Makassar 90245, Indonesia; 3Research Institute for Humanity and Nature, 457-4 Motoyama, Kamigamo, Kita-ku, Kyoto 603-8047, Japan; kimijima@chikyu.ac.jp; 4Faculty of Collaborative Regional Innovation, Ehime University, 3 Bunkyo-cho, Matsuyama 790-8577, Japan; 5Faculty of Medicine, Universitas Negeri Gorontalo, Jenderal Sudirman Street No. 6, Gorontalo 96100, Indonesia; 6Cyclotron Research Center, Iwate Medical University, 348-58 Tomegamori, Takizawa 020-0173, Japan; ksera@iwate-med.ac.jp

**Keywords:** heavy metal contamination, river sediments, ASGM, Gorontalo, geo-accumulation index

## Abstract

Substances found in watersheds and sediments in artisanal and small-scale gold mining (ASGM) areas contaminated by heavy metals are becoming tremendously critical issues in Asia. This study aimed at clarifying the pollution caused by heavy metals in sediments in river basins near ASGM sites in Gorontalo Province, North Sulawesi, Indonesia. Sediment samples collected from experimental areas were classified into nine clay samples and twenty-seven sand samples, whereas three other samples were collected from the control area. Particle-induced X-ray emission was used to analyze these samples. The Statistical Package for the Social Science and the geo-accumulation index (I_geo_) were also used for analysis. Based on the results, Hg, Pb, As, and Zn had a concentration of 0–334 µg/g, 5.5–1930 µg/g, 0–18,900 µg/g, and 0–4923.2 µg/g, respectively, which exceeded limits recommended by the U.S. Environmental Protection Agency consensus (1991) and the Indonesian Government Regulation Number 38, 2011. Furthermore, I_geo_ showed the order of the pollution degree Hg < Zn < Pb < As and reflected an environment contaminated by heavy metals, ranging from unpolluted to extremely polluted areas. Therefore, sediments contaminated by Hg, Pb, As, and Zn could be found along the river basin of mining areas.

## 1. Introduction

Heavy metal pollution from artisanal and small-scale gold mining (ASGM) has been a pivotal issue in Asian countries, with heavy metals found in watersheds and sediments. Along with many volcanic areas, Gorontalo has many ASGM sites with a unique process that is influenced by hydrothermal circulation and the type of geological setting. One of the main causes is mercury (Hg) pollution due to ASGM, which uses Hg in the amalgamation process to extract gold from ore rock [1,2].

Recent investigations conducted by the United Nations Environment Program highlighted the enormity of Hg pollution in developing countries and its harmful effects on human health and ecosystems [3,4,5,6,7]. Over 300,000 ASGM miners work at ~1000 informal sites in Indonesia [8,9,10,11,12]. ASGM presents an income opportunity for those in rural communities [13,14,15]; therefore, farmers or fishermen with few options in terms of alternative livelihoods become miners.

In Indonesia, heavy metal pollution tends to increase because of the high use of chemical substances. Since the industrialization era, the use of Hg in this sector has polluted industrial mine minerals, and one of the causes is that tailings resulting from the amalgamation process can lead to environmental hazards. ASGM located in some areas in Indonesia, particularly in Gorontalo, is a trigger of environmental damage. Moreover, volcanoes, animal waste, and algae as well as wood from deforestation can harm the environment [8,12,14,15].

Furthermore, heavy metal pollution from natural and anthropogenic activities is frequently detected in sediments and water columns of rivers, severely contaminating a large percentage of Asia’s rivers. This has become a major concern worldwide because most people lack access to clean drinking water and do not have adequate sanitation services [3,16]. In addition, heavy metals are easy to accumulate in sediments since the concentration of heavy metals is always higher in sediments than in river water [16]. Heavy metal content in sediments probably fluctuates due to the occurrence of a large undercurrent, which uplifts or displaces sediments to other locations [17,18,19,20,21,22]. Previous studies conducted using water and sediment samples from Bone River and Wubudu river of Gorontalo Province in Indonesia have revealed that these rivers have been polluted by Hg along with As and Pb due to activities at ASGM sites [19,20,21]. In contrast, this study particularly focused on major river sampling sites to clarify the effect of heavy metal pollution from sediments in river basins near ASGM sites in Gorontalo Province.

## 2. Materials and Methods

### 2.1. Research Areas

Gorontalo is a province in Indonesia with minerals and natural resources, particularly gold. Gold mining in Gorontalo Province has emerged as a good source of income with the potential to boost the economy and social welfare. This research was conducted in Buladu River near Hulawa Mining; Totopo River near Bumela Mining; Mopuya Daa River near Dunggilata Mining; Bone and Bula Rivers near Suwawa Mining; and Ayidu River (serving as a control area). These are parts of Gorontalo Province, Indonesia [9,12].

### 2.2. Samples

In this study, samples were collected based on the distance between the mining site and river, as data on Hg content in river water sediments were compared with data collected from the river closest to ASGM areas. Meanwhile, data on Hg levels in sediment samples from river water were collected from factory tailing ponds and amalgamation processing units, which are close to rivers and where mining processes occur.

Sediment samples were collected from five ASGM areas (Hulawa Mining, Bumela Mining, Dunggilata Mining, and Suwawa Mining) and one control area from November 2018 to September 2019. Nine clay samples and twenty-seven sand samples were collected (Figure 1).

Collected samples ranged from coarse sand to clay materials (approximately 2–0.002 mm). All samples were preserved in autoclaved sample bags. In addition, the samples were homogenized by grinding in an agate mortar to obtain fine particles [23,24,25].

### 2.3. Analytical Methods

In an experimental laboratory, sediment samples were dried in an oven for 48 h at 80–120 °C. Each dry sample was taken from the oven at 25% of its total volume and then filtered through a sieve to separate large-size materials, such as root, stem, or tree, which were <5 mm or 200–450 µm in size. Each sediment sample was crushed using a planetary micro mill, with jar settings of 5 min at speed 3, 3 min at speed 5, and 3 min at speed 9. In the subsequent stage, powder samples were transferred into a sample bottle using a spoon, inserted into sterile, sealed in plastic bottles, and homogenized at the facility for 60 min. Each sediment sample was measured to 50 mg and mixed with 10 mg of palladium carbon by using an agate mortar. Powder samples were added to a 3 µL collodion solution (collodion ethanol = 1:9) and then spread out (flat; homogeneous) on a thin film using a pipette tip. Concentrations of heavy metals were measured by particle-induced X-ray emission at the Cyclotron Research Centre, Iwate Medical University, Japan [24].

## 3. Results

### 3.1. Concentration of Hg in River Sediments at ASGM Area and Control Area

The Hg concentration originated from mining (ASGM) sites and the control area (rivers flowing from upstream to downstream). On determination of each element in heavy metal concentration and descriptive analysis in sand and clay of sediment sample at sampling point in Gorontalo Province, Indonesia, and IBM SPSS Statistic 24 for Windows (IBM Inc., Chicago, IL, USA) was applied for the standard established that is preferred for four heavy metals (Hg, Pb, As, and Zn) because these heavy metals are related to sediment and are included in each area of the ASGM site around the river sediment.

Table 1 shows that Totopo River has the highest median Hg concentration of clay (0.15 µg/g), and the other river has the lowest concentration (0 µg/g). On the other hand, the highest median Hg concentration of sand was found in Buladu River (54.7 µg/g), and the lowest (10.1 µg/g) was found in Dunggilata River.

In addition, some sand and clay samples were collected from the same mining sites, such as Sumalata Mining near Buladu River, Bumela Mining near Totopo River, Dunggilata Mining near Mopuya Daa River, and Suwawa Mining near Bone River, to compare the element concentration. In contrast, Hg content in the river sediment samples was very high, above the detection or threshold limit. Hg content (concentration) was above the detection limits of U.S. Environmental Protection Agency and river sediments of water quality criteria based on Indonesian Government Regulation Number 38, 2011 [26,27,28]. This was also in line with previous research that stated that Hg concentrations in Wubudu River sediments were far above the threshold limits stated by World Health Organization because the river is close to ASGM processing units [29,30].

### 3.2. Concentration of Pb in River Sediments at ASGM Area and Control Area

The Pb content originates from mining (ASGM) sites and the control area, with rivers flowing from upstream. Table 2 shows the results of the laboratory analysis in which the median concentration of Pb (µg/g) in sand sediment samples ranged from 93.6 to 237 µg/g, and clay sediment samples ranged from 36.3 to 1286.5 µg/g.

### 3.3. Concentrations of As in River Sediments at ASGM Area and Control Area

Table 3 shows results of the laboratory analysis in which the median concentration of As (µg/g) in sand sediment samples ranged from 0.8 to 315.1 µg/g, and that in clay sediment samples are 0 µg/g based on the interpretation of geo-accumulation index (I_geo_) classification. For As concentration, the result indicated clay sediment samples with a grain size of 0.0025 mm had high concentration compared to sand sediment samples taken from along the river and ASGM sites, especially in Sumalata Mining and Suwawa Mining of Gorontalo Province, Indonesia. Subsequently, the concentration of As is extremely polluted if compared to others concentration of heavy metals, such as Hg, Pb, and Zn [9,21]. In addition, arsenic (As) and iron (Fe) can be associated with the environment (natural source), especially in mineral clay. Moreover, arsenic is also one of the byproducts of the processing of non-ferrous metal ores, especially gold, which has very toxic properties with damaging effects on the environment. It is easy to find in several metal ore deposits, including Cu-Zn-Pb deposits containing enargite minerals and Cu-pyrite deposits, and interestingly [31,32].

### 3.4. Concentration of Zn in River Sediments at ASGM Area and Control Area

The Zn content originated from mining (ASGM) sites and the control area, with rivers flowing from upstream to downstream. Based on of results in Table 4, the median concentration of Zn (µg/g) in sand sediment samples ranged from 60.4 to 187.5 µg/g, and clay sediment samples ranged from 3.2 to 2562.1 µg/g. In addition, zinc also an element of moderate abundance in the Earth’s crust. Surface sulfides liberate the soluble Zn^2+^ ion, which may form secondary carbonate and silicate minerals. Zinc sulfides are closely associated with Cd and As, and the extraction of metallic zinc is in some cases responsible for environmental increases in cadmium, copper, nickel, lead, and other heavy metals [33].

### 3.5. I_geo_ in Heavy Metal Concentration

I_geo_ was originally defined by Muller (1979) to determine and define metal contamination in sediments by comparing current metal concentrations with pre-industrial levels [34]. This method was used to determine levels of contamination or accumulation of heavy metals in sediments or soil (Table 5).
$$I_{\mathrm{geo}}\;={\;\log}_{2}(C_{i}/1.5\;B_{i})$$

$C_{i}$: The measured concentration of the examined metal (*n*) in sediment;

$B_{i}$: Geochemical background concentration of the metal (*n*).

Based on the Muller scale (1981), Hg I_geo_ values in the sand in study areas (Figure 2) belonged to I_geo_ ≤ 0 in Class 0 (46.1%), 0 < I_geo_ < 1 in Class 1 (15.3%), 1 < I_geo_ < 2 in Class 2 (11.5%), 2 < I_geo_ < 3 in Class 3 (19.2%), and 3 < I_geo_ < 4 in Class 4 (7.7%), reflecting an unpolluted to moderately polluted sand. They were classified as moderately polluted to strongly polluted sand (or I_geo_ values −0.7 to 3.1). Meanwhile, considering I_geo_, most clays (Figure 2) were included in Class 0 (I_geo_ ≤ 0) and Class 1 (0 < I_geo_ ≤ 1), i.e., unpolluted to moderately polluted clay, accounting for 44.4% and 11.1% of samples, respectively, and only clay samples in Class 6 (I_geo_ > 5) accounting for 44.4% of the samples. I_geo_ values in the sand in study areas (Figure 3) were from I_geo_ ≤ 0 in Class 0 (50%), 0 < I_geo_ < 1 in Class 1 (15.3%), 1 < I_geo_ < 2 in Class 2 (11.5%), 2 < I_geo_ < 3 in Class 3 (15.3%), 3 < I_geo_ < 4 in Class 4 (3.8%), and I_geo_ > 5 in Class 6 (3.8%), reflecting an unpolluted to moderately polluted sand, from moderately polluted to strongly polluted sand, and from strongly polluted to extremely polluted sand in the environment (or I_geo_ values −5.9 to 4.1). However, considering the I_geo_, most clays (Figure 3) were included in Class 0 (I_geo_ ≤ 0) and Class 1 (0 < I_geo_ ≤ 1), i.e., unpolluted to moderately polluted, with 55.5% and 11.1% of samples, respectively, and only some clays in Class 6 (I_geo_ > 5) with 33.3% of the samples. Pb I_geo_ values on the sand in study areas (Figure 4) were from I_geo_ ≤ 0 in Class 0 (26.9%), 0 < I_geo_ < 1 in Class 1 (23%), 1 < I_geo_ < 2 in Class 2 (23%), 2 < I_geo_ < 3 in Class 3 (19.2%), and 3 < I_geo_ < 4 in Class 4 (7.6%), reflecting an unpolluted to moderately polluted sand and from moderately polluted to strongly polluted sand in the environment (or I_geo_ values −3.6 to 3.1). In contrast, considering I_geo_, most of the clay samples (Figure 4) were included in Class 0 (I_geo_ ≤ 0) and Class 1 (0 < I_geo_ ≤ 1), that is, unpolluted to moderately polluted clay, accounting for 44.4% and 11.1% of samples, respectively, and only some clays in Class 6 (I_geo_ > 5) accounted for 44.4% of the samples. Pb I_geo_ values on the sand in the study areas (Figure 4) were from I_geo_ ≤ 0 in Class 0 (26.9%), 0 < I_geo_ < 1 in Class 1 (23%), 1 < I_geo_ < 2 in Class 2 (23%), 2 < I_geo_ < 3 in Class 3 (19.2%), and 3 < I_geo_ < 4 in Class 4 (7.6%), reflecting an unpolluted to moderately polluted sand, and from moderately polluted to strongly polluted sand in the environment (or I_geo_ values of −3.6–3.1). Subsequently, considering I_geo_, most clay samples (Figure 4) were included in Class 0 (I_geo_ ≤ 0) and Class 1 (0 < I_geo_ ≤1), i.e., unpolluted to moderately polluted clay, accounting for 44.4% and 11.1% of samples, respectively, and only some clays in Class 6 (I_geo_ > 5) accounted for 44.4% of samples [35,36].

Zn I_geo_ values for the sand in study areas (Figure 5) were from I_geo_ ≤ 0 in Class 0 (26.9%), 0 < I_geo_ < 1 in Class 1 (7.6%), 1 < I_geo_ < 2 in Class 2 (38.4%), 2 < I_geo_ < 3 in Class 3 (23%), and 3 < I_geo_ < 4 in Class 4 (3.8%), reflecting an unpolluted to moderately polluted sand, and from moderately polluted to strongly polluted sand in the environment (or I_geo_ values of −4.9–3.9). Subsequently, considering I_geo_, most clay samples (Figure 5) were included in Class 0 (I_geo_ ≤ 0) and Class 1 (0 < I_geo_ ≤ 1), i.e., unpolluted to moderately polluted clay, accounting for 55.5% and 11.1% of samples, respectively, and only some clay samples in Class 6 (I_geo_ > 5) accounted for 33.3% of samples.

Based on the results, distribution of heavy metals in the research areas varied from upstream to downstream. Generally, content of heavy metals was relatively higher in upstream areas than downstream areas [37].

**Table 5 ijerph-19-06094-t005:** I_geo_ by Muller’s classification for geochemical index [34,38,39,40,41,42,43].

I_geo_ Value	Class	Quality of Sediment
≤0	0	Unpolluted
0–1	1	From unpolluted to moderately polluted
1–2	2	Moderately polluted
2–3	3	From moderately to strongly polluted
3–4	4	Strongly polluted
4–5	5	From strongly to extremely polluted
>5	6	Extremely polluted

### 3.6. Correlation Data of Each Element

Hg and As elements are positively correlated with a correlation coefficient of 0.82 (82%), represented by graph C of Figure 6, whereas the others, namely are graph A where Pb correlated with As with a correlation coefficient of 0.58 (58%) and graph B where Pb correlated with Hg with a correlation coefficient of 0.66 (66%) in Figure 6, are categorized into moderately correlated. This occurred because the community tends to add excessive amounts of Hg to strengthen the bound gold during the gold mining process. In contrast, addition of excess Hg causes tailing waste in the environment to be higher [44,45].

## 4. Discussion

### 4.1. Sources of Heavy Metals in the River Sediment of Gorontalo Province

Considering I_geo_, some sediments were classified as Class 6 (I_geo_ > 5), indicating that they are extremely polluted, and only a few sediments were classified as Class 0 (I_geo_ ≤ 0), Class 4 (3 < I_geo_ < 4), and Class 5 (4 < I_geo_ < 5), indicating they ranged from unpolluted to strongly polluted and from strongly polluted to extremely polluted in the environment polluted by Hg, Pb, As, and Zn.

Hg, Pb, As, and Zn elements contribute to pollution in air, water, sediments, and soils. Their contributions would negatively impact the environment on Hg contributions. It is an untypical heavy metal, as it can completely contaminate an environment, which occurs during gold mining, in which Hg can be released into the air and can also be widely distributed through tailings, especially on ASGM sites. Meanwhile, Pb and As can only be found in other heavy metals found in water, sediment, and soil [34,36,37,43,46].

### 4.2. The Effect of Heavy Metal Pollution on Sediments

We measured the concentration of heavy metal pollution in sediment samples collected from two field locations (ASGM area and control area) along the river in Gorontalo Province, Indonesia (Table 6). The ASGM area comprises five rivers that are still active with human activities in gold mining sites, and it also has many trommel houses and six gold mining sites. The control area is located along a highway with no mining activity. The river basin was the determining factor in changing the lower degree of chemical weathering as well as the climate from dry and cold to slightly warmer and more humid. Low chemical weathering could have influenced the lower degree of ion exchange from rock weathering to river solute in the watersheds of Gorontalo Province, Indonesia [20,21,22]. However, relative rivers imply that warm and humid climate conditions prevailed in their watersheds. Thus, it releases a significant number of major ions from silicate weathering into rivers [44].

### 4.3. Impact of ASGM on/for River Sediment

People in Gorontalo Province have been mining for gold since 1970. They used the traditional method, where the processing process does not use high technology and uses very simple equipment instead. Gold processing is conducted in several stages, such as rock excavation, processing unit, and waste disposal. Each stage of this process has an ecological impact that can disturb the environmental balance; hence, it must be handled carefully to reduce the risk of environmental damage. In general, almost every mining disposal area in Gorontalo Province is a land from which waste flows into the river, so metals and other materials accumulate in the waste, influencing the river ecosystem. As an ecosystem, rivers are targets for waste disposal with high levels of pollution affecting the life of the aquatic biota. It has been argued that water is often polluted by inorganic components, including dangerous heavy metals [36].

The use of these heavy metals in everyday life (directly or indirectly) intentionally, unintentionally, or intentionally but indirectly, has polluted the environment, where certain types of pollution have polluted the environment beyond the threshold for human life. Pollutants such as Hg are metals accumulated in the body of an organism and will stay in the body for a long time as a poison [31].

Traditional gold mining is one of the economic activities in the community from which miners earn a living. This activity has been a source of pollution to the surrounding environment due to the use of simple technologies such as using Hg as a binder of the gold element in the amalgamation process. Pollution occurs when Hg used as a binding agent for the gold element is discharged with washing wastewater to disposal sites both on the ground and in river water [22].

As a result, these traditional mines (ASGM) with various activities causing transportation of substances in sediments starting upstream is harmful when heavy metals are accumulated downstream because they can pollute the environment and endanger humans in the area.

Therefore, the source of mercury pollution in gold processing in Gorontalo Province for this study occurred from gold ore processing activities with the amalgamation process. In the amalgamation process, gold is separated from the binder where the gold ore that has been in the form of fine granules is an amalgamation process. Mercury will automatically bind to gold. Tailings or mining wastes from the amalgamation process, which contain a great deal of mercury, are directly discharged into the environment (rivers) without being processed first, so it is very possible to cause water pollution, especially in rivers at the gold mining location. This is what causes mercury (Hg) to have a strong relationship between leads (Pb) and arsenic (As) [33].

In addition, based on BLHRD Gorontalo Province in 2015 and 2018, the community around the river of ASGM site directly disposes of household waste and gold processing into the river. As a result, there is a decrease in river water quality because biochemical oxygen demand (BOD) levels become increased in that condition [47].

### 4.4. Heavy Metal Interaction (Hg, Pb, and As) Release in the Environment from Source of ASGM Site to River Sediment in this Study Area

As seen in Figure 6A–C, the element that has the strongest relationship is Hg with As, which is 0.82 (82%). Then, the second is the Hg element with Pb of 0.66 (66%), while the weakest relationship level is found in the Pb element with As of 0.58 (58%). This is based on that in the gold mining process, as the community tends to add excessive amounts of Hg with the intention to bind more to gold. On the other hand, the addition of excess Hg causes the amount of Hg that is wasted as pollution into the environment to also be higher if the gold content contained in the ore is not as large as expected.

Mercury is a heavy metal element that is widely used by humans in industrial or mining processes. In nature, mercury is a metal that can be found in two forms: organic and inorganic. Organic mercury is a mercury compound that binds to the carbon element to form methyl, ethyl, or similar functional groups, while inorganic mercury includes mercury vapor (Hg_0_), mercury salt (Hg^2+^), and mercury metal. Organic mercury is a geochemical form of mercury that has the highest level of toxicity to humans [4]. The traditional gold mining process is the largest anthropogenic contributor to mercury pollution, and one of the gold ore extraction methods widely used by traditional gold mining is amalgamation [20,21]. In the gold mining process, people tend to add excess mercury so that there are larger amounts of bound gold. The addition of excess mercury causes the mercury that is wasted as pollution in the environment to be higher. The gold content contained in the ore is often not as large as the community expects, so an experiment in adding mercury is expected with maximum results [9].

In addition, mercury pollution also occurs in the tailings process. Tailings are technically defined as fine materials that are minerals that are left after valuable minerals are extracted in an ore processing [10].

Tailings are also considered to be pulp that no longer contains valuable minerals, but given that processing costs do not reach 100%, it is still possible to have gold in the tailings [18]. Gold mining tailings contain one or more toxic hazardous materials, such as arsenic (As), cadmium (Cd), lead (Pb), mercury (Hg), and cyanide (Sn). Tailings consist of crushed rock derived from mineral rock that has been mined of minerals. Tailings can be in the form of solids such as very fine sand or slurry, namely solid tailings mixed with water to form a thin layer [1,18].

In general, tailings disposal is carried out in a terrestrial environment that is in a topographic depression or artificial reservoir, river or lake, and sea. Tailings often contain valuable mineral concentrations that do not meet the requirements for collection at the time of mining but are stored for future use. Mineralogically tailings can consist of various minerals such as silica, iron silicate, magnesium, sodium, potassium, and sulfide. Of these minerals, sulfides have chemically active properties, and when in contact with air, they will experience oxidation to form acidic salts and acidic streams containing a number of toxic metals, such as, Hg, Pb, and Cd [11].

Therefore, the source of mercury pollution in gold processing in Gorontalo Province for this study occurred from gold ore processing activities with the amalgamation process. In the amalgamation process, gold is separated from the binder where the gold ore that has been in the form of fine granules. Mercury will automatically bind to gold. Tailings or mining wastes from the amalgamation process, which contain a great deal of mercury, are directly discharged into the environment (rivers) without being processed first, so it is very possible that this causes water pollution, especially in rivers at the gold mining location. This is what causes mercury (Hg) to have a strong relationship between leads (Pb) and arsenic (As) [19,20,21].

## 5. Conclusions

In general, gold mining in Gorontalo Province is one of the potential natural resources that provide better prospects in an improved-level economy and social welfare. This economic improvement is mainly in terms of income, employment, and opportunities for new activities outside the agriculture sector plantation [12].

Despite the negative side, artisanal mining plays an essential role in developing societies. Small mines can be a major source of revenue for rural communities and can provide income for investment. Artisanal miners can exploit mineral deposits considered uneconomic by modern industry. Every USD 1 generated through artisanal mining generates about USD 3 in non-mining jobs. In Indonesia, artisanal mining is very useful as a means of livelihood for poor people and has been proven as a safety net in time of economic distress, especially during the economic crisis occurred in 1998, which lasted for about 10 years [15].

Moreover, on the basis of the interaction correlation analysis between sand and clay through a computerized analysis statistic, Pb and As, Pb and Hg, and Hg and As had a positive correlation with correlation coefficients of 0.58, 0.66, and 0.82, respectively. In addition, most ASGM sites in Gorontalo have hydrothermal processes that influence the geological setting, making them important sources of As in the environment. In addition, I_geo_ proposed by Muller in 1969 was used to evaluate the degree of heavy metal pollution in the river sediments [38,43]. It revealed that sediments along the river basin were severely polluted by heavy metals, and their estimated I_geo_ values were in the following increasing order: Hg < Zn < Pb < As. Regarding the Muller scale in 1981, I_geo_ was evaluated for all sampling sites of all five rivers in the Gorontalo Province. Of the sampling sites, 90% had I_geo_ greater than 5, which means they should be considered as extremely polluted. However, heavy metal values analyzed by using the I_geo_ indicator in Gorontalo Province were relatively high, indicating severe pollution by Hg, Pb, As, and Pb.

Furthermore, river sediments collected along river mining sites are contaminated with As, which is the highest among other heavy metals found in Gorontalo compared to any region in Indonesia or any country in the world.

## Figures and Tables

**Figure 1 ijerph-19-06094-f001:**
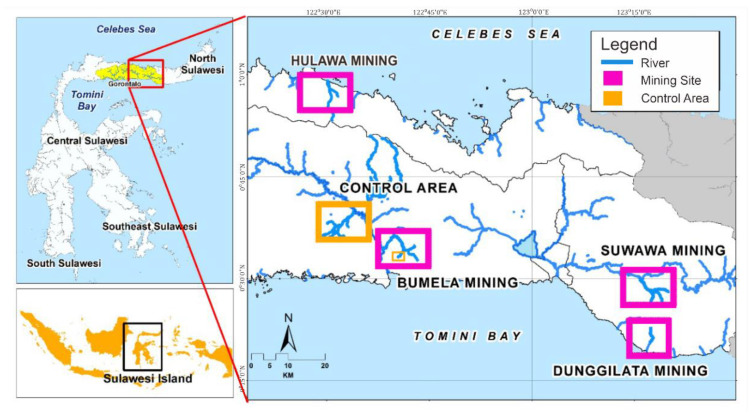
Map of the location of the research area.

**Figure 2 ijerph-19-06094-f002:**
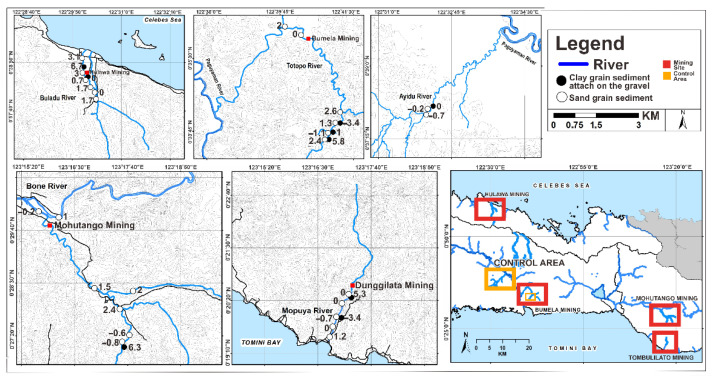
Plots of I_geo_ and contaminant levels in Hg along the river sediments in Gorontalo Province, Indonesia (µg/g).

**Figure 3 ijerph-19-06094-f003:**
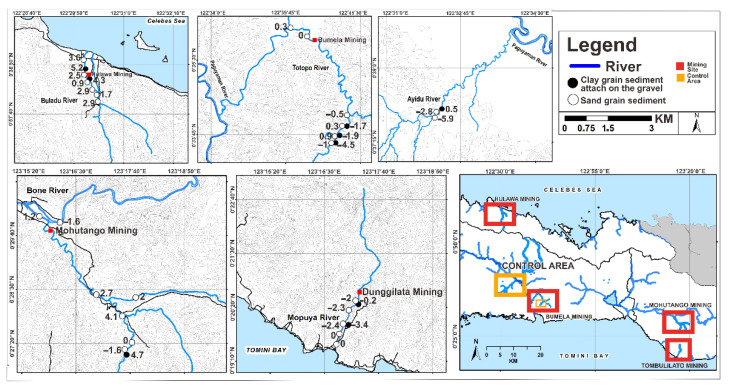
Plots of I_geo_ and contaminant levels in As along the river sediments in Gorontalo Province, Indonesia (µg/g).

**Figure 4 ijerph-19-06094-f004:**
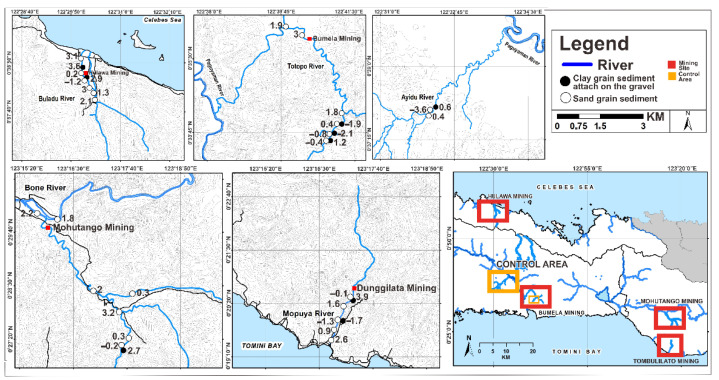
Plots of I_geo_ and contaminant levels in Pb along the river sediments in Gorontalo Province, Indonesia (µg/g).

**Figure 5 ijerph-19-06094-f005:**
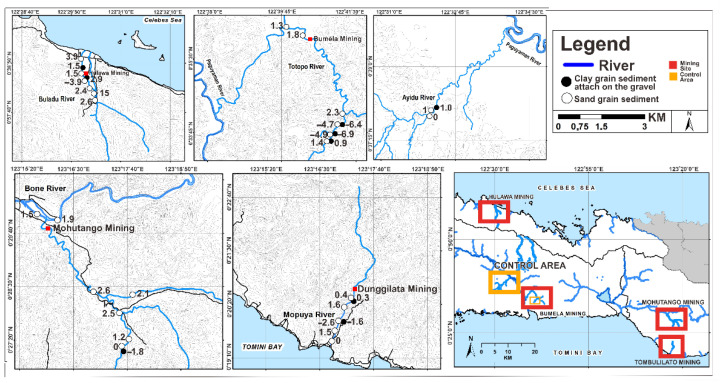
Plots of I_geo_ and contaminant levels in Zn along the river sediments in Gorontalo Province, Indonesia (µg/g).

**Figure 6 ijerph-19-06094-f006:**
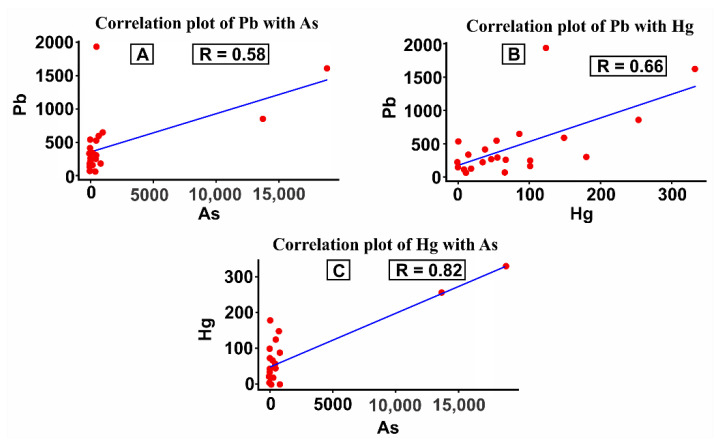
(**A**) Correlation between Pb and As; (**B**) Correlation between Pb, and Hg; (**C**) Correlation between Hg, and As in Gorontalo Area.

**Table 1 ijerph-19-06094-t001:** Concentration of Hg (µg/g) in river sediment sample from ASGM and control area.

Location	Element	Median	Min–Max
Buladu River (Sumalata Mining)	Sand	54.7	0–147
	Clay	0	0–334
Dunggilata River (Dunggilata Mining)	Sand	10.1	0–38.1
	Clay	0	0–124
Totopo River (Bumela Mining)	Sand	53.6	0–101
	Clay	0.15	0–179
Bone River (Suwawa Mining)	Sand	33.7	86.3–9.5
	Clay	0	0–256
Ayidu River (Control area)	Sand	14.9	10.5–101
	Clay	0	0–0

**Table 2 ijerph-19-06094-t002:** Concentration of Pb (µg/g) in river sediment sample from ASGM and control area.

Location	Element	Median	Min–Max
Buladu River (Sumalata Mining)	Sand	230.5	30.2–589
	Clay	1286.5	0–1610
Dunggilata River (Dunggilata Mining)	Sand	124	27–412
	Clay	985.2	0–1930
Totopo River (Bumela Mining)	Sand	168.9	38.1–536
	Clay	36.3	0–308
Bone River (Suwawa Mining)	Sand	237	58.2–645
	Clay	856	0–856
Ayidu River (Control area)	Sand	93.6	5.5–187
	Clay	198	0–198

**Table 3 ijerph-19-06094-t003:** Concentration of As (µg/g) in river sediment sample from ASGM and control area.

Location	Element	Median	Min–Max
Buladu River (Sumalata Mining)	Sand	315.1	0–587
	Clay	0	0–18,900
Dunggilata River (Dunggilata Mining)	Sand	9.2	0–11.9
	Clay	0	0–449
Totopo River (Bumela Mining)	Sand	46.3	0–88.5
	Clay	11.3	0–162
Bone River (Suwawa Mining)	Sand	108	15.9–798
	Clay	0	0–13,800
Ayidu River (Control area)	Sand	0.8	0–6.7
	Clay	0	0–729

**Table 4 ijerph-19-06094-t004:** Concentration of Zn (µg/g) in river sediment sample from ASGM and control area.

Location	Element	Median	Min–Max
Buladu River (Sumalata Mining)	Sand	187.5	3.2–673.7
	Clay	2562.1	0–4923
Dunggilata River (Dunggilata Mining)	Sand	60.4	0–138.1
	Clay	210.1	0–331
Totopo River (Bumela Mining)	Sand	114.7	1.5–234.6
	Clay	3.2	0–153.5
Bone River (Suwawa Mining)	Sand	175.1	0–282.4
	Clay	79.4	0–79.4
Ayidu River (Control area)	Sand	90.8	0–102
	Clay	549.2	0–549.2

**Table 6 ijerph-19-06094-t006:** I_geo_ by Using Average Data in Control Area (Ayidu River and Totopo River) [19,20,21].

Classification of River Sediment Quality Standard	Element Concentration Limit			
	As	Pb	Hg	Zn
Clay Grain Sediment Attach on Gravel	342.3	88.5	2.1	184.9
Sand Grain Sediment	32	45.7	11.1	30.8
Average Data in Control Area (Ayidu River and Totopo River)				

## Data Availability

Not applicable.

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
