# Peer review of "Contamination Level in Geo-Accumulation Index of River Sediments at Artisanal and Small-Scale Gold Mining Area in Gorontalo Province, Indonesia"

_ijerph, 2022, doi:10.3390/ijerph19106094_

Round 1
Reviewer 1 Report
After reading carefully the article " Contamination Level in Geo-accumulation Index of River Sediments at Artisanal and Small-scale Gold Mining Area in Gorontalo Province, Indonesia ”, I believe that the authors of the manuscript appropriately corrected the earlier version of the article. The authors supplemented the results (subsections 3.1, 3.3, and 3.4). First of all, the discussion is supplemented and conducted much better, similarly to the conclusions. The article contains interesting content, I believe that it can be published in a corrected version in the journal.
Reviewer 2 Report
I am satisfied with the answers of the authors and the corrections they have made to the text
This manuscript is a resubmission of an earlier submission. The following is a list of the peer review reports and author responses from that submission.
Round 1
Reviewer 1 Report
Overall, the text is confusing.
In the introduction, in particular the last paragraph, should be revised.
Section 3 should be rewritten as it is difficult to follow the classification into classes of contamination by the various metals. In section 3.6, the correlation coefficients between metals have values of R=0.58 for Pb and As, R=0.66 for Pb and Hg and R=0.82 for Hg and As, these R values are low and, furthermore, if we analyze the graphs, we can see that only 3 points were considered for the calculation of R, in a much larger universe of samples, which makes the meaning of the correlation invalid. I think that both the discussion part and the conclusions should be redone based on a new evaluation of the results.
Author Response
Dear Editor of IJERPH
On behalf of all the authors, I am pleased to re-submit our revision of manuscript with title “Contamination Level in Geo-Accumulation Index of River Sediments at Artisanal and Small-Scale Gold Mining Area in Gorontalo Province, Indonesia” for publication in IJERPH Journal as an original research article.
We greatly appreciate the reviewers for their complimentary comments and suggestions. We have carried out the extensive revision that the reviewers suggested. Please find attached a point-by-point response to reviewer’s concerns.
I realize that many sentences require English editing. After reaching the minor revision, we will send our manuscript to the scientific and English editing company.
Thank you for your consideration!
Sincerely yours,
Basir
Public Health Faculty
Hasanuddin University

Reviewer 2 Report
Manuscript entitled „Contamination Level in Geo-accumulation Index of River Sediments at Artisanal and Small-scale Gold Mining Area in Gorontalo Province, Indonesia” it is interesting and fits into a journal International Journal of Environmental Research and Public Health. The discussion and conclusion chapters needs to be supplemented and developed
Discusion 244
4.2. The Effect of Heavy Metal Pollution on Sediments
There is no reference to literature
4.3. Impact of ASGM on/for River Sediment
The negative impact of heavy metal contaminated sediments on the natural environment of Gorontalo Province should be much more emphasized
Conclusion
This is rather a summary, the conclusions should be changed, what are the ecological and economic effects of this phenomenon, you can try to outline the directions of activities in the future
Author Response

(The authors gave the same response as above.)

Reviewer 3 Report
The article is devoted to the actual problem - the influence of gold mining on the chemical composition of bottom sediments. The authors analyzed 36 samples of bottom sediments for the content of a number of metals and metalloids. The use of Igeo made it possible to characterize the risk of contamination of bottom sediments in comparison with the background content of metals and metalloids.
The authors used modern methods of research and processing of the obtained results. Nevertheless, I have several questions and comments to the authors.
- The authors use the term “heavy metals” for Hg, Pb, As, and Zn, but As refers to metalloid
- Fig. 1. It is not clear which places are up and down the ASGM Area.
- Table 1. The content of Hg at the control site in the sand does not differ from other sites of the study. Was statistical analysis performed to determine if there were significant differences between control and ASGM Area? Such an analysis must be done for all elements. The authors compare the content in sediments with the standards. It is advisable to indicate these standards in tables 1, 2, 3, 4. For what type of bottom sediments are the standards established?
- Table. 2. How to explain that the Pb content in Totopo River (Bumela Mining,? Content in clay is lower than in the control? Is there a statistically significant difference between these values?
- Table. 3. The As content in the control clay is higher than in Totopo River (Bumela Mining), Dunggilata River (Dunggilata Mining). Why?
- Table. 4. Similar questions
- Igeo calculation uses background values. How did the authors calculate the index? Were the values from the control point used for each type of sediment (separately for sand and clay)? In some cases, the background content was higher than in the ASGM Area.
- Figures 2-5 do not illustrate the results well. It is better to present data in the form of graphs rather than maps.
- Fig. 6. Pair correlations of As-Pb and As-Hg are not convincing, because most of the points at the origin of the axis can be considered as one point.
- Discussion. A better discussion of the results is required.
- L.239-242. This conclusion is not illustrated by figures or tables, data are not given above and below the mining sites.
- L.295. Muller in 1969 - no citation, L.298 Muller in 1981 - no citation
General conclusion. The article may be interesting for readers, but significant revision is required.
Author Response

(The authors gave the same response as above.)

Round 2
Reviewer 1 Report
I believe that the manuscript has been improved enough to warrant publication in IJERPH, it only needs minor English corrections, like the authors said.
Author Response

(The authors gave the same response as above.)

Reviewer 3 Report
After the revision of the article, I still have some questions and comments to the authors
- Table 1. The content of Hg at the control site in the sand does not differ from other sites of the study.
Regarding Hg no reply received. Why is the control content in sand higher than in mining areas? It is necessary to indicate in tables 1-4 or in the text the standards with which the authors compare the content of metals in sediments
- 2. How to explain that the Pb content in Totopo River (Bumela Mining) сontent in clay is lower than in the control?
The explanation is unsatisfactory. In my opinion, the Totopo River average value cannot be used due to the big dispersion in the data, as indicated by the SD. This concerns all metals in every investigated place. What is the reason for such heterogeneity of data?
- Igeo calculation uses background values. If the average values for some metals in the control are higher than in mining places, how can Igeo be calculated? In this case, it is more appropriate to use not the average values, but the median. The median must also be placed in the table. 1-4
- Figures 2-5. The figures proposed by the authors are more visual and appropriate than the previous version.
Author Response

(The authors gave the same response as above.)
